# DI-MaskDINO: A Joint Object Detection and Instance Segmentation Model

Zhixiong Nan[1], Xianghong Li[1], Tao Xiang [*][1], and Jifeng Dai[2]

[1]College of Computer Science, Chongqing University, Chongqing, China.
[2]Department of Electronic Engineering, Tsinghua University, Beijing, China.
`nanzx@cqu.edu.cn, lixianghong@stu.cqu.edu.cn, txiang@cqu.edu.cn,`
`daijifeng@tsinghua.edu.cn`

## Abstract

This paper is motivated by an interesting phenomenon: the performance of object detection lags behind that of instance segmentation (i.e., *performance imbalance*) when investigating the intermediate results from the beginning transformer decoder layer of MaskDINO (i.e., the SOTA model for joint detection and segmentation). This phenomenon inspires us to think about a question: will the *performance imbalance* at the beginning layer of transformer decoder constrain the upper bound of the final performance? With this question in mind, we further conduct qualitative and quantitative pre-experiments, which validate the negative impact of *detection-segmentation imbalance* issue on the model performance. To address this issue, this paper proposes **DI-MaskDINO** model, the core idea of which is to improve the final performance by alleviating the *detection-segmentation imbalance*. **DI-MaskDINO** is implemented by configuring our proposed ***De-Imbalance (DI)*** module and ***Balance-Aware Tokens Optimization (BATO)*** module to MaskDINO. ***DI*** is responsible for generating balance-aware query, and ***BATO*** uses the balance-aware query to guide the optimization of the initial feature tokens. The balance-aware query and optimized feature tokens are respectively taken as the *Query* and *Key&Value* of transformer decoder to perform joint object detection and instance segmentation. **DI-MaskDINO** outperforms existing joint object detection and instance segmentation models on COCO and BDD100K benchmarks, achieving **+1.2** $AP^{box}$ and **+0.9** $AP^{mask}$ improvements compared to SOTA joint detection and segmentation model MaskDINO. In addition, **DI-MaskDINO** also obtains **+1.0** $AP^{box}$ improvement compared to SOTA object detection model DINO and **+3.0** $AP^{mask}$ improvement compared to SOTA segmentation model Mask2Former.

## 1 Introduction

Object detection and instance segmentation are two fundamental tasks in the computer vision community. Intuitively, the two tasks are closely-related and mutually-beneficial. However, in the current time, specialized detection or segmentation gains more focuses, and the amount of works studying the specialized task is significantly larger than that for joint tasks. One typical explanatory is that the multi-task training even hurts the performance of the individual task.

Confronting current research situations, we think about whether there are some essential cruxes that are ignored in previous works and these cruxes hinder the cooperation of object detection and instance

---

[*]Corresponding author.

This work is supported by Chongqing Natural Science Foundation Innovation and Development Joint Fund (CSTB2023NSCQ-LZX0109).

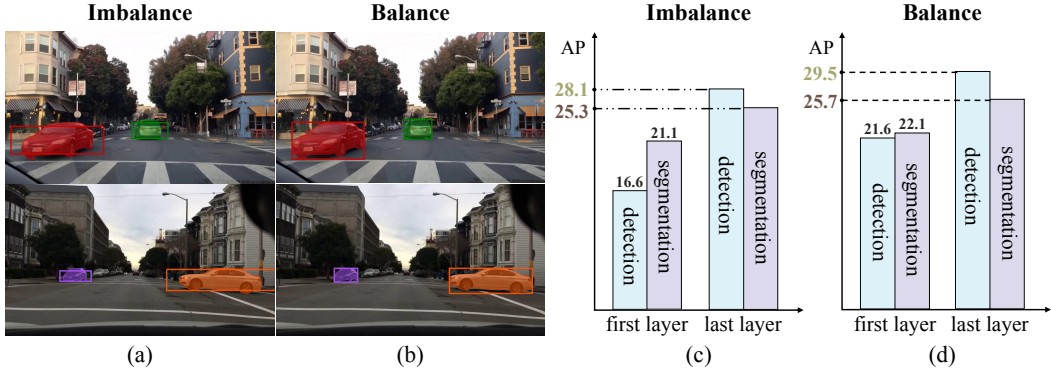

Figure 1: Qualitatively, (a) shows that the detection bounding boxes predicted by the query/feature at the first decoder layer of MaskDINO do not fit well with segmentation masks, and (b) exhibits that the corresponding results of **DI-MaskDINO** are optimized and the detection bounding boxes closely surround segmentation masks. Quantitatively, (c) displays that there exists a significant performance gap between detection and segmentation at the first decoder layer of MaskDINO, and (d) demonstrates **DI-MaskDINO** not only alleviates the performance imbalance at the first layer but also improves the performance upper bound.

segmentation tasks, which further constrains the breakthrough of the performance upper bound. This paper reveals one of cruxes is the imbalance of object detection and instance segmentation. As shown in Fig. 1, when investigating the results at the first layer of transformer decoder of MaskDINO model [25], an interesting phenomenon is found that there exists the performance imbalance between object detection and instance segmentation, as qualitatively illustrated in Fig. 1a and quantitatively illustrated in the first bar of Fig. 1c. After considering the imbalance issue, the performance gap at the first layer is narrowed as illustrated in the first bar of Fig. 1d, and the final performance upper bounds are improved (i.e., 28.1 to 29.5 for detection and 25.3 to 25.7 for segmentation) as illustrated in the second bar of Fig. 1d. The qualitative results are also optimized, the detection bounding boxes closely surround segmentation masks, as illustrated in Fig. 1b.

According to the above analysis, we can find the detection-segmentation imbalance at the beginning layer is one of essential cruxes that hinders the cooperation of object detection and instance segmentation. Therefore, we reviewed the previous works to investigate whether there are works that have been aware of this issue. The idea of many classical and excellent methods [16, 2, 5, 11] is combining two tasks by adding a segmentation branch to an object detector. These detect-then-segment methods make the performance of segmentation task to be limited by the performance of the object detector. Thanks to the thriving of transformer [44] and DETR [3], recent research attention has been geared towards transformer-based methods, which make giant contributions to the community. For example, [10, 50, 25] use the unified query representation for object detection and instance segmentation tasks based on transformer architecture.

However, to our best knowledge, there is no existing work to solve the above mentioned detection-segmentation imbalance issue. Factually, the imbalance issue naturally exists, which is determined by the **individual characteristics** of detection and segmentation tasks and also derived from the **supervision manners** for the two tasks. **Firstly**, segmentation is a pixel-level grouping and classification task [16, 46], thus local detailed information is important for this task. In contrast, detection is a region-level task to locate and regress the object bounding box [13, 38], which requires global information focusing on the complete object. The query at the beginning decoder layer conveys relatively local features, which is more beneficial for the segmentation task, thus the detection task tends to achieve lower performance at the beginning layer. **Secondly**, supervision manners for detection and segmentation are distinctly different. The segmentation is densely supervised by all pixels of the GT mask, while detection is sparsely supervised by a 4D vector (i.e., x, y, w, and h) of GT bounding box. The dense supervision provides richer and stronger information than the sparse supervision during the optimization procedure. Therefore, the optimization speeds of the two tasks are not synchronous, which will lead to the imbalance issue.

Based on two above-analyzed reasons for the detection-segmentation imbalance issue, it is straightforward that the performance of detection task will lag behind at the beginning layer. Considering existing

methods share a unified query for detection and segmentation tasks, the performance of a task will be negatively affected by another poorly-performed task, leading to that the multi-task joint training even hurts the performance of the individual task. Therefore, addressing the detection-segmentation imbalance issue is significant for designing a joint object detection and instance segmentation model. To address the detection-segmentation imbalance issue, we propose **DI-MaskDINO** model, which is implemented by configuring our proposed ***De-Imbalance (DI)*** module and ***Balance-Aware Tokens Optimization (BATO)*** module to MaskDINO. ***DI*** module is responsible for generating balance-aware query. Specifically, ***DI*** module strengthens the detection at the beginning decoder layer to balance the performance of the two tasks, and the core of ***DI*** module is our proposed ***residual double-selection*** mechanism. In this mechanism, the *token interaction* based ***double-selection*** structure learns the global geometric, contextual, and semantic patch-to-patch relations to update initial feature tokens, and high-confidence tokens are selected to benefit the detection task since the selected tokens have learned global semantics during the *token interaction* procedure. In addition, this mechanism makes use of initial feature tokens by the ***residual*** structure, which is the necessary compensation for the information loss occurring during ***double-selection***. Apart from ***DI*** module, we also design ***BATO*** module, which uses the balance-aware query to guide the optimization of initial feature tokens. The balance-aware query and optimized feature tokens are respectively taken as the *Query* and *Key&Value* of transformer decoder to perform joint object detection and instance segmentation.

The contributions of this paper are as follows: ***i***) to our best knowledge, this paper for the first time focuses on the detection-segmentation imbalance issue and proposes ***DI*** module with the ***residual double-selection*** mechanism to alleviate the imbalance; ***ii***) **DI-MaskDINO** outperforms existing SOTA joint object detection and instance segmentation model MaskDINO (**+1.2** $AP^{box}$ and **+0.9** $AP^{mask}$ on COCO, using ResNet50 backbone with 12 training epochs), SOTA object detection model DINO (**+1.0** $AP^{box}$ on COCO), and SOTA segmentation model Mask2Former(**+3.0** $AP^{mask}$ on COCO).

## 2   Related Work

**Object Detection.** Classical object detection methods [13, 38, 27, 37, 5, 43, 42] have achieved significant success. In recent years, transformer-based methods such as DETR [3] make a giant contribution to object detection by introducing the concept of object queries and the one-to-one matching mechanism. The success of DETR has sparked a boom in query-based end-to-end detectors, and numerous excellent variants are proposed [56, 33, 52, 28, 23, 6, 47, 51, 24, 32, 21, 54, 18]. Specifically, to enhance the convergence speed of DETR, Deformable DETR [56] proposes deformable attention mechanism that learns sparse feature sampling and aggregates multi-scale features accelerating model convergence and improving performance. From the interpretability of object queries, DAB-DETR [28] formulates the queries as 4D anchor boxes and dynamically updates them in each decoder layer.

**Instance Segmentation.** CNN-based instance segmentation methods are categorized into top-down and bottom-up paradigms. The top-down paradigm [16, 2, 11, 19, 7, 4, 1] firstly generates bounding boxes by object detectors, and then segments the masks. The bottom-up paradigm [35, 9, 29, 12] treats instance segmentation as a labeling-clustering problem, where pixels are firstly labeled as a class or embedded into a feature space and then clustered into each object. Recently, many transformer-based instance segmentation methods [8, 14, 20, 53, 17, 10, 50, 25] are proposed. The transformer-based methods treat the instance segmentation task as a mask classification problem that associates the instance categories with a set of predicted binary masks.

**Joint Object Detection and Instance Segmentation.** The goal of joint object detection and instance segmentation is to carry out the two tasks simultaneously [45, 34, 41, 36]. The traditional joint object detection and instance segmentation methods [16, 2, 5, 11] are usually implemented by adding a mask branch to a strong object detector. For example, the classical model Mask RCNN [16] achieves joint object detection and instance segmentation by adding a mask branch to Faster RCNN [39]. Recently, the proposal of the transformer-based framework has promoted the development of joint object detection and instance segmentation. Following DETR [3], SOLQ [10] proposes a unified query representation for simultaneous detection and segmentation tasks. The recent MaskDINO [25] achieves optimal performance with the unified query representation on both detection and segmentation tasks.

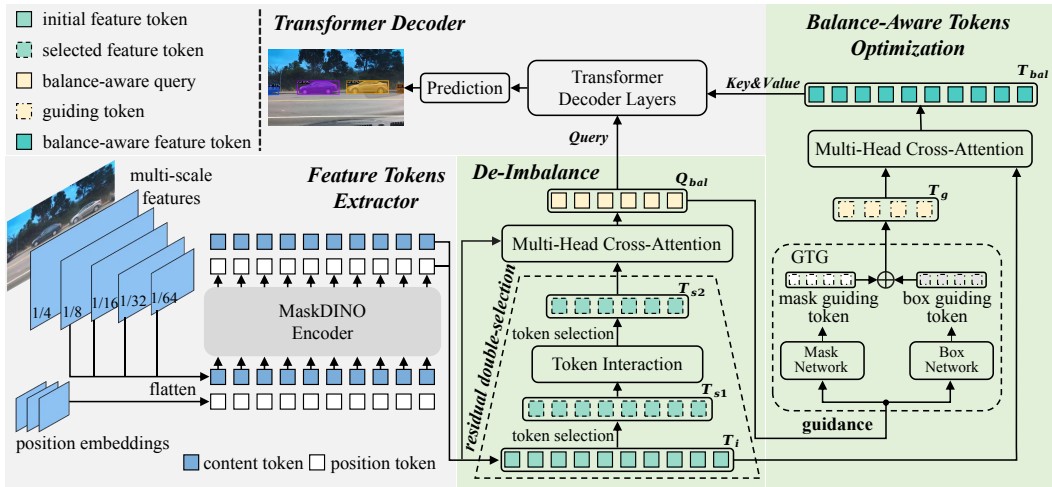

Figure 2: The overview of **DI-MaskDINO** model based on MaskDINO (grey shaded), with the extensions (green shaded) of *De-Imbalance* and *Balance-Aware Tokens Optimization*. For simplicity, content token and position token are merged in *De-Imbalance* (i.e., $T_i$, $T_{s1}$, $T_{s2}$, and $Q_{bal}$ contain both content and position token) in presentation. GTG is short for guiding token generation.

## 3 Proposed Method

In response to the naturally-existing but commonly-ignored imbalance issue between object detection and instance segmentation, we propose **DI-MaskDINO** model, which is based on MaskDINO [25]. To better understand our proposed **DI-MaskDINO**, we firstly review MaskDINO (§ 3.1), and then introduce **DI-MaskDINO** (§ 3.2).

### 3.1 Preliminaries: MaskDINO

MaskDINO is a unified object detection and segmentation framework, which adds a mask prediction branch on the structure of DINO [52]. MaskDINO (grey shaded part in Fig. 2) is composed of a backbone, a transformer encoder, a transformer decoder, and detection and segmentation heads (i.e., "Prediction" in Fig. 2). Position embeddings and the flattened multi-scale features (extracted by backbone) are inputted to the transformer encoder to generate the initial feature tokens ($T_i$). Note that in MaskDINO, the top-ranked feature tokens selected from $T_i$ directly serve as the *Query* of transformer decoder, while we design *De-Imbalance* module with a *residual double-selection* mechanism to firstly alleviate the detection-segmentation imbalance and then obtain the balance-aware query $Q_{bal}$ to serve as the *Query* of transformer decoder. In addition, we design *Balance-Aware Tokens Optimization* module to optimize $T_i$ and generate the balance-aware feature tokens $T_{bal}$ to serve as the *Key&Value* of transformer decoder. Token and query are specialized terms, and their explanations are provided in Appendix A.

### 3.2 Our Method: DI-MaskDINO

Fig. 2 illustrates the overview of **DI-MaskDINO**, which consists of four modules, including *Feature Tokens Extractor* (*FTE*), *De-Imbalance* (*DI*), *Balance-Aware Tokens Optimization* (*BATO*), and *Transformer Decoder* (*TD*). *FTE* extracts the initial feature tokens $T_i$ from the input image using backbone and MaskDINO encoder. The goal of *DI* is to generate the balance-aware query $Q_{bal}$, which is implemented by applying our proposed *residual double-selection* mechanism on the initial feature tokens $T_i$. *BATO* utilizes $Q_{bal}$ to optimize $T_i$, generating the balance-aware feature tokens $T_{bal}$. *TD* takes $T_{bal}$ as the *Key&Value* and $Q_{bal}$ as the *Query* to perform joint object detection and instance segmentation.

### 3.2.1 Feature Tokens Extractor

Given an image $I \in \mathbb{R}^{H \times W \times 3}$, the backbone (e.g., ResNet [15]) firstly extracts multi-scale features, which are then flattened and concatenated to serve as the input of transformer encoder comprising six multi-scale deformable attention layers [56], obtaining the initial feature tokens $T_i$ that is composed of $\sum_{i=3}^{6} \left( \frac{H}{2^i} \times \frac{W}{2^i} \right)$ tokens, where each token expresses the feature of the corresponding patch in $I$.

### 3.2.2 De-Imbalance

There exists the detection-segmentation imbalance at the beginning layer of transformer decoder, which might constrain the upper bound of model performance. To handle this issue, we design ***DI*** module to alleviate the imbalance, instead of directly providing $T_i$ to the transformer decoder as MaskDINO does. Specifically, detection-segmentation imbalance means that the performance of object detection lags behind that of instance segmentation at the beginning layer of transformer decoder. Therefore, we propose the ***residual double-selection*** mechanism to strengthen the performance of object detection.

The ***double-selection*** consists of the first selection and the second selection. In the first selection, we select top-$k_1$ ranked feature tokens in $T_i$, based on their category classification scores:

$$T_{s1} = \mathcal{S}(T_i, k_1), \tag{1}$$

where $T_{s1}$ represents the firstly-selected feature tokens, $\mathcal{S}$ denotes the selection operator. The first selection mainly filters out most of the tokens conveying background information, making $T_{s1}$ focus on the objects.

Before the second selection, a *token interaction* network comprising two self-attention layers is applied on $T_{s1}$:

$$T_{s1}^{sa} = \text{MHSA}(T_{s1}), \tag{2}$$

where MHSA is Multi-Head Self-Attention and $T_{s1}^{sa}$ indicates the feature tokens processed by MHSA.

The *token interaction* is the key point to make sure that the secondly-selected tokens are beneficial for detection, we explain its rationality as follows. As we know, each token actually corresponds to a patch (remarkably smaller than an object in most cases) in the image [55]. The bounding box of an object is regressed by integrating the multiple patches (belonging to the same object) that have global patch-to-patch spatial relations, thus it is really needed for the detection task to learn the interaction relation between patches. In contrast, the dense all-pixel supervision for the segmentation task mainly focuses on local pixel-level similarity with GT mask [25], hence the segmentation task is not particularly depend on the patch-to-patch relation as the detection task. By self-attention layers, different tokens representing the patches (belonging to the same object) can interact with each other to learn the global geometric, contextual, and semantic patch-to-patch relations, benefiting the perception of object bounding boxes. Therefore, executing *token interaction* before the second selection makes ***DI*** module to be more beneficial for detection. In addition, verified by some studies (e.g., [32]), the tokens with higher category scores correspond to higher IOU scores of object bounding boxes. Therefore, the second selection further strengthens the object detection and alleviates the detection-segmentation imbalance.

In the second selection, we select the top-$k_2$ ranked feature tokens in $T_{s1}^{sa}$ to obtain the secondly-selected feature tokens $T_{s2}$:

$$T_{s2} = \mathcal{S}(T_{s1}^{sa}, k_2). \tag{3}$$

The ***residual double-selection*** is inspired by the ***residual*** idea in [15], and the ***residual*** is the necessary compensation for ***double-selection*** since the information loss occurs in the selection procedures. The formulation of this mechanism is combining $T_i$ with $T_{s2}$ by the Multi-Head Cross-Attention network (MHCA, a self-attention layer and a FFN layer are omitted here), generating $Q_{bal}$:

$$Q_{bal} = \text{MHCA}(T_{s2}, T_i). \tag{4}$$

$Q_{bal}$ conveys the balance-aware information, thus it is named as balance-aware query. Subsequently, $Q_{bal}$ is fed to ***BATO*** to guide the optimization of initial feature tokens $T_i$. It is noted that the tokens in $Q_{bal}$ have become significantly different from the tokens in $T_i$. Through Eq. 1-4, the tokens in $Q_{bal}$ have obtained larger receptive field and considered the interaction with other tokens, thus they are better understood as object/instance candidates. Correspondingly, the denotation has been changed from $T$ to $Q$.

### 3.2.3 Balance-Aware Tokens Optimization

In MaskDINO, initial feature tokens $\boldsymbol{T}_i$ directly serve as the *Key&Value* of $\boldsymbol{TD}$. Instead, we design $\boldsymbol{BATO}$ module that makes use of both balance-aware query $\boldsymbol{Q}_{bal}$ and $\boldsymbol{T}_i$ to generate the *Key&Value* of $\boldsymbol{TD}$. $\boldsymbol{T}_i$ contains a large number ($\approx$ 20k) of tokens conveying detailed local information for both background and objects/instances, while $\boldsymbol{Q}_{bal}$ consists of a small number (=300) of high-confidence tokens mainly focusing on objects/instances. In addition, benefiting from the *token interaction* (i.e., Eq. 2), $\boldsymbol{Q}_{bal}$ has learned rich semantic and contextual interaction relations. Therefore, $\boldsymbol{Q}_{bal}$ is used to guide the optimization of $\boldsymbol{T}_i$. The optimized feature tokens (denoted as $\boldsymbol{T}_{bal}$) is taken as the *Key&Value* of $\boldsymbol{TD}$.

Firstly, to provide guidance for both detection and segmentation, the mask network and box network are separately applied on $\boldsymbol{Q}_{bal}$ to generate mask guiding token $\boldsymbol{T}_g^{mask}$ and box guiding token $\boldsymbol{T}_g^{box}$:

$$\boldsymbol{T}_g^{mask} = \mathcal{N}_{mask}(\boldsymbol{Q}_{bal}), \tag{5}$$

$$\boldsymbol{T}_g^{box} = \mathcal{N}_{box}(\boldsymbol{Q}_{bal}), \tag{6}$$

where $\mathcal{N}_{mask}$ and $\mathcal{N}_{box}$ indicate the mask network and box network, respectively. Both $\mathcal{N}_{mask}$ and $\mathcal{N}_{box}$ consist of a $mlp$ network.

Then, the overall guiding token $\boldsymbol{T}_g$ is obtained by fusing $\boldsymbol{T}_g^{mask}$ and $\boldsymbol{T}_g^{box}$ :

$$\boldsymbol{T}_g = \boldsymbol{T}_g^{mask} + \boldsymbol{T}_g^{box}. \tag{7}$$

Finally, $\boldsymbol{T}_g$ guides the optimization of the initial feature tokens $\boldsymbol{T}_i$ through a Multi-Head Cross-Attention. The motivation is straightforward. Same with $\boldsymbol{Q}_{bal}$, each token in $\boldsymbol{T}_g$ corresponds to an object/instance candidate. When $\boldsymbol{T}_i$ interacts with $\boldsymbol{T}_g$, the tokens (in $\boldsymbol{T}_i$) that belong to the same object/instance will be aggregated, enhancing the foreground information. For a better comprehension, a token in $\boldsymbol{T}_g$ could be assumed as the center of a "cluster", and the tokens (in $\boldsymbol{T}_i$) belonging to the same object/instance could be assumed as the points in the "cluster". The points move towards the "cluster" center, realizing the optimization of $\boldsymbol{T}_i$. This procedure is formulated as follows:

$$\boldsymbol{T}_{bal} = \mathrm{MHCA}(\boldsymbol{T}_i, \boldsymbol{T}_g), \tag{8}$$

generating balance-aware feature tokens $\boldsymbol{T}_{bal}$ (also called optimized feature tokens), which serve as the *Key&Value* of $\boldsymbol{TD}$.

### 3.2.4 Transformer Decoder

$\boldsymbol{TD}$ is responsible for the predictions of instance mask, object box, and class. $\boldsymbol{TD}$ consists of decoder layers and each layer contains a self-attention, a cross-attention, and a FFN. The inputs of $\boldsymbol{TD}$ are $\boldsymbol{T}_{bal}$ (in Eq. 8) and $\boldsymbol{Q}_{bal}$ (in Eq. 4). $\boldsymbol{Q}_{bal}$ interacts with $\boldsymbol{T}_{bal}$ in the decoder layers, continuously refining the query:

$$\boldsymbol{Q}_{ref} = \mathcal{N}_{de}(\boldsymbol{Q}_{bal}, \boldsymbol{T}_{bal}), \tag{9}$$

where $\boldsymbol{Q}_{ref}$ is the refined query, and $\mathcal{N}_{de}$ denotes the transformer decoder network.

Subsequently, we follow the detection head and segmentation head structures of MaskDINO to perform object detection and instance segmentation. For object detection, $\boldsymbol{Q}_{ref}$ is used to predict the categories $\boldsymbol{c}$ and bounding boxes $\boldsymbol{b}$:

$$\{\boldsymbol{c}, \boldsymbol{b}\} = \mathcal{N}_{det}(\boldsymbol{Q}_{ref}), \tag{10}$$

where $\mathcal{N}_{det}$ denotes the detection head network. For instance segmentation, $\boldsymbol{Q}_{ref}$, $\boldsymbol{T}_i$, and the 1/4 resolution CNN feature $\boldsymbol{F}_{cnn}$ are used to predict the instance masks $\boldsymbol{m}$:

$$\boldsymbol{m} = \mathcal{N}_{seg}(\boldsymbol{Q}_{ref}, \boldsymbol{T}_i, \boldsymbol{F}_{cnn}), \tag{11}$$

where $\mathcal{N}_{seg}$ denotes the segmentation head network.

## 4 Experiments

### 4.1 Settings

We conduct extensive experiments on COCO [26] and BDD100K [49] datasets using ResNet50 [15] backbone pretrained on ImageNet-1k [40] as well as SwinL [30] backbone pretrained on ImageNet-22k. NVIDIA RTX3090 GPUs are used when the backbone is ResNet50. Due to the large memory

requirement of SwinL, NVIDIA RTX A6000 GPUs are used when the backbone is SwinL. More implementation details are in Appendix B.

Table 1: Comparison with other methods on the COCO validation set.

| Methods | Epochs | $AP^{box}$ | $AP_S^{box}$ | $AP_M^{box}$ | $AP_L^{box}$ | $AP^{mask}$ | $AP_S^{mask}$ | $AP_M^{mask}$ | $AP_L^{mask}$ | FPS |
|---|---|---|---|---|---|---|---|---|---|---|
| **ResNet50 backbone** | | | | | | | | | | |
| Mask RCNN [16] | 36 | 41.0 | 24.9 | 43.9 | 53.3 | 37.2 | 18.6 | 39.5 | 53.3 | 21.0 |
| HTC [5] | 36 | 44.9 | - | - | - | 39.7 | 22.6 | 42.2 | 50.6 | 5.5 |
| SOLQ [10] | 50 | 48.5 | 30.1 | 51.6 | 64.8 | 40.1 | 20.9 | 43.7 | 59.4 | 7.0 |
| DINO [52] | 36 | 50.9 | 34.6 | 54.1 | 64.6 | - | - | - | - | 6.8 |
| Mask2Former [8] | 50 | - | - | - | - | 43.7 | 23.4 | 47.2 | 64.8 | 5.3 |
| MaskDINO [25] | 12 | 45.7 | - | - | - | 41.4 | 21.1 | 44.2 | 61.4 | 8.0 |
| DI-MaskDINO (Ours) | 12 | **46.9 (+1.2)** | 28.8 | 49.5 | 62.9 | **42.3 (+0.9)** | 22 | 44.8 | 62.8 | 7.7 |
| MaskDINO [25] | 24 | 48.4 | - | - | - | 44.2 | 23.9 | 47.0 | 64.0 | 8.0 |
| DI-MaskDINO (Ours) | 24 | **49.6 (+1.2)** | 31.7 | 52.6 | 65.3 | **44.8 (+0.6)** | 24.3 | 47.8 | 65.0 | 7.7 |
| MaskDINO [25] | 50 | 51.7 | 34.2 | 54.7 | 67.3 | 46.3 | 26.1 | 49.3 | 66.1 | 8.0 |
| DI-MaskDINO (Ours) | 50 | **51.9 (+0.2)** | 36.3 | 54.7 | 66.7 | **46.7 (+0.4)** | 27.5 | 49.8 | 66.2 | 7.7 |
| **SwinL backbone** | | | | | | | | | | |
| MaskDINO [25] | 12 | 52.2 | 34.8 | 55.6 | 69.9 | 47.2 | 26.3 | 50.3 | 69.1 | 3.4 |
| DI-MaskDINO (Ours) | 12 | **53.3 (+1.1)** | 36.7 | 56.7 | 70.4 | **47.9 (+0.7)** | 27.7 | 51.1 | 69.3 | 3.0 |
| MaskDINO [25] | 50 | 56.8 | 40.2 | 60.2 | 72.3 | 51.0 | 31.3 | 54.1 | 71.2 | 3.4 |
| DI-MaskDINO (Ours) | 50 | **57.8 (+1.0)** | 41.5 | 61.2 | 73.9 | **51.8 (+0.8)** | 31.8 | 55.1 | 72.2 | 3.0 |

Table 2: Comparison with other methods on the BDD100K validation set.

| Methods | Epochs | $AP^{box}$ | $AP_S^{box}$ | $AP_M^{box}$ | $AP_L^{box}$ | $AP^{mask}$ | $AP_S^{mask}$ | $AP_M^{mask}$ | $AP_L^{mask}$ | FPS |
|---|---|---|---|---|---|---|---|---|---|---|
| **ResNet50 backbone** | | | | | | | | | | |
| Mask RCNN [16] | 50 | 25.5 | 15.7 | 32.8 | 56.1 | 20.7 | 15.7 | 26.6 | 49.1 | 20.1 |
| HTC [5] | 50 | 26.9 | 15.7 | 35.4 | 55.3 | 21.1 | 11.0 | 26.7 | 46.4 | 4.8 |
| SOLQ [10] | 50 | 27.0 | 16.5 | 35.3 | 45.6 | 19.6 | 10.1 | 25.8 | 37.4 | 6.3 |
| DINO [52] | 36 | 28.9 | 18.0 | 37.0 | 48.1 | - | - | - | - | 6.1 |
| Mask2Former [8] | 50 | - | - | - | - | 19.6 | 8.4 | 25.9 | 41.0 | 4.7 |
| MaskDINO [25] | 68 | 28.1 | 17.4 | 36.1 | 47.9 | 25.3 | 14.2 | 31.8 | 48.1 | 6.7 |
| DI-MaskDINO (Ours) | 68 | **29.5 (+1.4)** | 18.0 | 37.4 | 50.4 | **25.7 (+0.4)** | 14.5 | 32.1 | 48.1 | 6.4 |
| **SwinL backbone** | | | | | | | | | | |
| MaskDINO | 68 | 30.2 | 19.0 | 37.5 | 48.6 | 27.0 | 15.4 | 32.6 | 50.5 | 3.2 |
| DI-MaskDINO (Ours) | 68 | **31.4 (+1.2)** | 19.4 | 40.4 | 48.7 | **27.9 (+0.9)** | 16.6 | 34.1 | 51.2 | 2.8 |

## 4.2 Comparison Experiments

MaskDINO is the SOTA model for joint object detection and instance segmentation, thus we mainly compare our model with MaskDINO under different backbones (ResNet50 and SwinL). Additionally, our model is compared with some classical (i.e., Mask RCNN [16]) and recently-proposed (i.e., HTC [5] and SOLQ [10]) joint object detection and instance segmentation models. Furthermore, our model is compared with SOTA model that is specifically designed for object detection (i.e., DINO [52]) and instance segmentation (i.e., Mask2Former [8]). The comparison results on COCO dataset are summarized in Tab. 1. It is noted that the experiments with the Swin-L backbone are conducted on the A6000 GPUs with the batch size of 4 (the maximum bacth size that 4 A6000 GPUs supports). The batch size is smaller than that in MaskDINO paper (i.e., batch size = 16) and the 4 A6000 GPUs present weaker computation power than 4 A100 GPUs, thus the results we reproduced are lower than those in the original MaskDINO paper. We can observe that **1)** our model surpasses MaskDINO under different training conditions (epoch = 12, 24, and 50). Notably, our model presents more significant advantage with the training condition of epoch = 12, which potentially reveals that our model reaches the convergence with a faster speed; **2)** under the Swin-L backbone, **DI-MaskDINO** exhibits significant superiority over MaskDINO, further confirming the effectiveness of our model; **3)** the performance of our model on individual detection and segmentation tasks is simultaneously higher than that of SOTA models specifically designed for detection (i.e., DINO) and segmentation (i.e., Mask2Former) when they are fully trained (epoch = 36 or 50), which is really hard-won since the single-task model usually designs the specific module for the specific task (e.g., DINO uses the tailored query formulation to improve the detection performance and Mask2Former proposes tailored masked attention module to improve the segmentation performance).

Existing joint detection and segmentation models like [5, 10, 50] only conduct the experiments on COCO dataset. In this paper, to further verify the robustness and generalization of our model, additional experiments are conducted on more complex traffic scene dataset BDD100K [49] using ResNet50 and Swin-L backbones, and the results we reproduce are shown in Tab. 2. Due to the complexity of traffic scenes, the overall performance is lower than the performance on COCO dataset, and the model asks for more training epochs (epoch = 68) to reach the convergence. It can be observed that our model still exhibits superiority over MaskDINO, DINO, and Mask2Former, which presents the robustness and generalization of our model. It should be noted that the performance of MaskDINO on detection task is lower than that of the specialized object detection model DINO, indicating that DINO still exhibits the advantage in complex traffic scene datasets. In contrast, our model improves DINO by 0.6 $AP^{box}$, further demonstrating the effectiveness of our model.

### 4.3 Diagnostic Experiments

#### 4.3.1 Imbalance Tolerance Test

There exists the natural imbalance between object detection and instance segmentation, and we are interested in how will a model perform if the imbalance condition is worsened. Therefore, we conduct the imbalance tolerance test by designing two severe imbalance conditions: **1)** *loss weight constraint*, which is implemented by constraining the weight of detection loss to 1/10 of the default value while the weight of segmentation loss remains unchanged; **2)** *position token constraint*, position token conveys important cues of object locations, thus constraining position token will generate disturbing location information to confuse detection task. The position token constraint is implemented by randomly initializing position token of $Q_{bal}$ (composed of position token and content token) in Eq. 4. ***DI*** module is mainly responsible for alleviating imbalance issue, thus the imbalance tolerance test on **DI-MaskDINO** only enables ***DI*** module. The experiments are conducted on more complex BDD100K dataset, because the results on the more complex dataset can better reflect the performance of imbalance tolerance. Considering the imbalance issue is severe at the beginning decoder layer, thus the experiments utilize models configured with 3 decoder layers.

Table 3: Imbalance tolerance test comparison of MaskDINO and **DI-MaskDINO**.

| Imbalance conditions | MaskDINO | | DI-MaskDINO (Ours) | |
|---|---|---|---|---|
| | $AP^{box}$ | $AP^{mask}$ | $AP^{box}$ | $AP^{mask}$ |
| standard | 27.5 | 23.7 | 27.9 | 24.9 |
| loss weight constraint | 24.7 (**-10.2%**) | 23 (**-3.0%**) | 27.1 (**-2.9%**) | 25.2 (**+1.2%**) |
| position token constraint | 21.5 (**-21.8%**) | 21.9 (**-7.6%**) | 23.8 (**-14.7%**) | 23.7 (**-4.8%**) |

The results of imbalance tolerance test are summarized in Tab. 3, and the percentage of performance drops (compared with standard condition) is highlighted in colors. We can observe that **1)** the imbalance between detection and segmentation has remarkable affect on the upper bound of model performance, potentially indicating the significance of our work; **2)** the effects of imbalance conditions on detection task are larger than that on segmentation task, because the two imbalance conditions are implemented to mainly constrain the detection task to simulate the natural detection-segmentation imbalance; **3)** even the performance of SOTA model MaskDINO is largely affected by the imbalance conditions (e.g., **-21.8%** $AP^{box}$ degradation on the condition of position token constraint), which potentially reflects that ***De-Imbalance*** deserves the research focus; **4)** compared with MaskDINO, the performance degradation of our model is slighter (i.e., **-10.2%** v.s. **-2.9%** and **-21.8%** v.s. **-14.7%** on the $AP^{box}$ metric, **-3.0%** v.s. **+1.2%** and **-7.6%** v.s. **-4.8%** on the $AP^{mask}$ metric), which demonstrates the effectiveness of our model; **5)** from a comprehensive perspective, we think the standard condition is still a detection-segmentation imbalance condition (which is commonly treated as a regular condition in previous works), and we claim the imbalance is one of the cruxes that limit the upper bound of model performance, hence it should be further studied.

#### 4.3.2 Diagnostic Experiments on Main Modules

To test the effects of main modules in our model (i.e., ***DI*** and ***BATO***), we test the performance of our model under four configurations: **#1** exclusion of both ***DI*** and ***BATO***; **#2** exclusion of ***BATO***; **#3** exclusion of ***DI***; **#4** inclusion of both ***DI*** and ***BATO***. To make the results solid, the experiments are conducted on both BDD100K and COCO datasets, and the results are reported in Tab. 4.

Table 4: The results of diagnostic experiments on main modules. The experiments are conducted on the BDD100K dataset with 68 training epochs and on COCO dataset with 12 training epochs. The results in Tab. 5 and Tab. 6 are also obtained under the same experiment settings.

| ID | *DI* | *BATO* | BDD100K | | COCO | |
|---|---|---|---|---|---|---|
| | | | $AP^{box}$ | $AP^{mask}$ | $AP^{box}$ | $AP^{mask}$ |
| **#1** | - | - | 27.8 | 24.4 | 45.6 | 41.2 |
| **#2** | ✓ | - | 28.8 | 25.2 | 46.4 | 42.1 |
| **#3** | - | ✓ | 28.3 | 24.9 | 46.2 | 41.8 |
| **#4** | ✓ | ✓ | **29.5** | **25.7** | **46.9** | **42.3** |

In comparison with **#1**, the model under the configuration of **#2** or **#3** yields higher performance on both datasets, and the optimum results are achieved when both *DI* and *BATO* are enabled (**#4**). These results demonstrate the effectiveness of *DI* and *BATO*. The results are explainable. *DI* module alleviates the imbalance between detection and segmentation, generating balance-aware query $Q_{bal}$, which is then fed to *BATO* to further make use of balance-aware information, contributing to performance improvement.

### 4.3.3 Diagnostic Experiments on *DI* Module

The core of our model is *DI*, which improves the model performance by mitigating the imbalance between detection and segmentation. *DI* is realized by applying the *residual double-selection* mechanism on $T_i$, generating $T_{s1}$ (firstly-selected tokens), $T_{s2}$ (secondly-selected tokens), and $Q_{bal}$ (balance-aware query). To analyze *DI* module, we design the fine-grained ablation experiments by respectively using $T_i$, $T_{s1}$, $T_{s2}$, and $Q_{bal}$ as the guidance for *BATO* (i.e., $gui. = T_i$, $gui. = T_{s1}$, $gui. = T_{s2}$, and $gui. = Q_{bal}$) and examine the corresponding performance.

The experiment results on BDD100K and COCO datasets are reported in Tab. 5. $gui. = T_i$ actually represents the situation when *DI* module is disabled, which serves as the baseline for other situations. Firstly, we can observe $\mathcal{P}(gui. = T_{s2}) > \mathcal{P}(gui. = T_{s1}) > \mathcal{P}(gui. = T_i)$ where $\mathcal{P}(*)$ denotes the performance of the model under the configuration $*$, demonstrating our *double-selection* mechanism is effective. The reason is intuitive, by applying *double-selection* mechanism, the tokens with high confidence are selected, and high-confidence tokens indicate the location of objects more clearly than other tokens, thus benefiting the object detection task (i.e., mitigating the imbalance be-

Table 5: The results of diagnostic experiments on *DI* module. $gui.$ denotes the **guidance** in Fig. 2.

| Guidance | BDD100K | | COCO | |
|---|---|---|---|---|
| | $AP^{box}$ | $AP^{mask}$ | $AP^{box}$ | $AP^{mask}$ |
| $gui. = T_i$ | 28.3 | 24.9 | 46.2 | 41.8 |
| $gui. = T_{s1}$ | 28.5 | 24.9 | 46.6 | 42.0 |
| $gui. = T_{s2}$ | 28.9 | 25.6 | 46.7 | 42.2 |
| $gui. = Q_{bal}$ | **29.5** | **25.7** | **46.9** | **42.3** |

tween detection and segmentation). Secondly, the highest performance is achieved when $gui. = Q_{bal}$, validating the effectiveness of our *residual double-selection* mechanism. In *DI* module, apart from the secondly-selected tokens $T_{s2}$, the initial feature tokens $T_i$ is also used to compute $Q_{bal}$, which could be coarsely formulated as $Q_{bal} = T_i + \mathcal{S}(T_i)$. This residual structure enables the model to make use of the information in both the initial feature tokens and the selected feature tokens, hence reaching the optimal performance.

### 4.3.4 Diagnostic Experiments on *BATO* Module

*BATO* targets to use the balance-aware query $Q_{bal}$ to guide the optimization of the initial feature tokens $T_i$. The effectiveness of *BATO* has been validated in Tab. 4. We further conduct experiments to validate the effect of the proposed guiding token generation (GTG). The GTG is designed to provide guidance for both detection and segmentation, generating mask guiding token and box guiding token that are closely related to mask instances and object boxes through the mask network and box network, respectively. These guiding tokens can provide more global and semantic guiding

Table 6: The results of diagnostic experiments on *BATO* module.

| Configurations | BDD100K | | COCO | |
|---|---|---|---|---|
| | $AP^{box}$ | $AP^{mask}$ | $AP^{box}$ | $AP^{mask}$ |
| w/o GTG | 28.6 | 25.4 | 46.5 | 42.2 |
| w/ GTG | **29.5** | **25.7** | **46.9** | **42.3** |

information for the optimization of the initial feature tokens $T_i$. As shown in Tab. 6, the model with GTG performs better, which demonstrates the effect of GTG.

## 5 Conclusion

In this paper, we investigate the naturally-existing but commonly-ignore detection-segmentation imbalance issue. The imbalance means that the performance of object detection lags behind that of instance segmentation at the beginning layer of transformer decoder, which is one of cruxes that hurt the cooperation of object detection and instance segmentation tasks and might constrain the breakthrough of the performance upper bound. To address the issue, we propose **DI-MaskDINO** model with the *residual double-selection* mechanism to alleviate the imbalance, achieving significant performance improvements compared with SOTA joint object detection and instance segmentation model MaskDINO, SOTA object detection model DINO, and SOTA segmentation model Mask2Former.

**Limitations.** This paper focuses on the task of joint object detection and instance segmentation, thus the model is not applicable for other segmentation tasks such as semantic segmentation and panoptic segmentation.

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

# Appendix

Due to the space limitation of the main text, we provide more results and discussions in the appendix, which are organized as follows:

## A  Prior Knowledge

**Encoder and decoder of DETR-like models.** MaskDINO is a type of DETR-like models [24, 54, 23, 47, 56, 51, 18]. A DETR-like model usually contains a backbone, an encoder, a decoder, and multiple prediction heads. The encoder is composed of multiple transformer encoder layers, and each encoder layer contains a multi-head self-attention and a FFN. The decoder consists of multiple transformer decoder layers, and each decoder layer has an extra multi-head cross-attention compared to the encoder layer.

**Token and query.** Both token and query are used to represent features. The concept of query comes from the original DETR [3]. Commonly, it denotes the feature of the object/instance in the decoder of DETR-like models. The token is a concept in the field of NLP (Natural Language Processing). In the computer vision domain, a token corresponds to a patch in an image. Feature tokens in this paper represent the features of image patches.

**How to obtain the intermediate results from the beginning transformer decoder layer.** The transformer decoder in MaskDINO is composed of multiple decoder layers, and MaskDINO attaches prediction heads after each decoder layer. Therefore, we can obtain prediction results from any decoder layer, which are called intermediate results. The intermediate results from the beginning transformer decoder layer are obtained by applying prediction heads on the 0-th decoder layer.

## B  Experiment Settings

### B.1  Datasets and Metrics

COCO [26] is the most widely used dataset for the object detection and instance segmentation tasks, and many well-known models such as [16, 5, 3, 52, 8, 25] are evaluated on the COCO dataset. Following the common practice, we use the COCO train2017 split (118k images) for training and the val2017 split (5k images) for validation. In addition, considering autonomous driving is a typical and practical application of object detection and instance segmentation, the experiments are also conducted on BDD100K [49] dataset, which is composed of 10k high-quality instance masks and bounding boxes annotations for 8 classes. The training set and validation set are divided following the standard in [49]. Consistent with previous researches [5, 10, 11, 50, 45, 25], we report the metrics of $AP^{box}$ and $AP^{mask}$ for performance evaluation.

### B.2  Implementation Details

We implement **DI-MaskDINO** based on Detectron2 [48], using AdamW [31] optimizer with a step learning rate schedule. The initial learning rate is set as 0.0001. Following MaskDINO, **DI-MaskDINO** is trained for 50 epochs on COCO with the batch size of 16, decaying the initial learning rate at fractions 0.9 and 0.95 of the total training iterations by a factor of 0.1. For BDD100K, following the setting in [22], we train our model for 68 epochs with the batch size of 8 and the learning rate drops at the 50-th epoch. The number of transformer encoder and decoder layers is 6.

The token numbers of $T_{s1}$ and $T_{s2}$ are 600 and 300, respectively. Unless otherwise specified, the feature channels in both encoder and decoder are set to 256, and the hidden dimension of FFN is set to 2048. The mask network and box network in **BATO** are both three-layer $mlp$ networks. We use the same loss function as MaskDINO (i.e., L1 loss and GIOU loss for box loss, focal loss for classification loss, and cross-entropy loss and dice loss for mask loss). Under ResNet50 pretrained on ImageNet [40], our model is trained on NVIDIA RTX3090 GPUs. For Swin-L backbone, NVIDIA RTX A6000 GPUs are used for training and validating.

## C   Additional Diagnostic Experiments

### C.1   Diagnostic Experiments on Token Selection

Token selection is a crucial step in our proposed ***residual double-selection*** mechanism. The number of token selection and the amount of selected tokens may affect the performance. We conduct experiments to verify their impacts. The experimental results are summarized in Tab. 7 and Tab. 8.

**The number of token selection.** Single-selection actually represents the situation when disabling ***DI*** module. ***Double-selection*** corresponds to our proposed method. Additionally, we add a token selection on $T_{s2}$ for triple-selection. For fair comparison, we set the amount of lastly-selected tokens to the same value (i.e., 300) for single-, double-, and triple-selection. From Tab. 7, we draw two observations: **1)** the results of single-selection are significantly lower than those in other situations, indicating the crucial role of ***DI*** module; **2)** our method achieves the optimal performance with ***double-selection***. The results are explainable. There exists information loss in each selection procedure. Therefore, triple-selection introduces more information loss, leading to a lower performance than ***double-selection***.

**The amount of selected tokens.** Three $k_1$ and $k_2$ settings in the ***double-selection*** mechanism are tested, and their maximum performance gap of $AP^{box}$ on BDD100K dataset is 0.4 (i.e., 29.5-29.1). Similar results are exhibited on the metric of $AP^{mask}$. These results demonstrate that our method is not sensitive to the hyper-parameters $k_1$ and $k_2$. Furthermore, the experiments on COCO dataset also exhibit the similar results, indicating that our method is robust. At last, we explain that the settings of $k_1$ and $k_2$ are infinite since they can be set as any value from 1 to 20k. SOTA models such as [18, 25, 24, 14] take 300 as the query number of transformer decoder. In the experiments, $k_1$ and $k_2$ are set as 300 or the multiple of 300 to align with the settings of the SOTA models.

Table 7: The results of diagnostic experiments on the number of token selection. Single-, double-, and triple-selection are represented as sing., doub., and trip., respectively. $k_i, i \in [1, 2, 3]$ denotes the amount of selected tokens.

| Datasets | Configurations | | $AP^{box}$ | $AP^{mask}$ |
|---|---|---|---|---|
| BDD100K | sing. | $k_1$=300 | 28.3 | 24.9 |
| | doub. | $k_1$=600,$k_2$=300 | **29.5** | **25.7** |
| | trip. | $k_1$=600,$k_2$=450,$k_3$=300 | 29.3 | 25.5 |
| COCO | sing. | $k_1$=300 | 46.2 | 41.8 |
| | doub. | $k_1$=600,$k_2$=300 | **46.9** | **42.3** |
| | trip. | $k_1$=600,$k_2$=450,$k_3$=300 | 46.6 | 42.2 |

Table 8: The results of diagnostic experiments on the amount of selected tokens with our proposed ***double-selection*** mechanism.

| Datasets | Configurations | $AP^{box}$ | $AP^{mask}$ |
|---|---|---|---|
| BDD100K | $k_1$=300,$k_2$=300 | 29.1 | 25.5 |
| | $k_1$=600,$k_2$=600 | 29.3 | 25.3 |
| | $k_1$=600,$k_2$=300 | 29.5 | 25.7 |
| COCO | $k_1$=300,$k_2$=300 | 46.6 | 42.3 |
| | $k_1$=600,$k_2$=600 | 46.8 | 42.4 |
| | $k_1$=600,$k_2$=300 | 46.9 | 42.3 |

### C.2   Diagnostic Experiments on the Number of Decoder Layer

We study the effect of different decoder layer numbers for the model performance, and the results are summarized in Tab. 9. We can observe the following: **1)** increasing the number of decoder layers

from 6 to 9 on BDD100K dataset results in the performance degradation, which can be explained by the inconsistency between the complexity of the model and the dataset. The BDD100K dataset only contains 7k training sets and 1k validation sets. The size of BDD100K is small and the model with 9 decoder layers is relatively more complex, leading to the overfitting of the training set; **2)** increasing the number of decoder layers will contribute to both detection and segmentation on COCO. However, the model configured with 9 decoder layers only achieves a slight improvement and introduces more computation cost. Therefore, we use 6 decoder layers in our model; **3)** our model has achieved comparable performance in the configuration with 3 decoder layers compared to MaskDINO with 9 decoder layers (e.g., 45.8 v.s. 45.7 on the $AP^{box}$ metric and 41.3 v.s. 41.4 on the $AP^{mask}$ metric on COCO), demonstrating that our model greatly improves the efficiency of the decoder.

Table 9: The results of diagnostic experiments on the number of decoder layer.

| Decoder layer | BDD100K | | COCO | |
| --- | --- | --- | --- | --- |
| | $AP^{box}$ | $AP^{mask}$ | $AP^{box}$ | $AP^{mask}$ |
| 3 | 28.7 | 25.3 | 45.8 | 41.3 |
| 6 | 29.5 | 25.7 | 46.9 | 42.3 |
| 9 | 28.9 | 25.6 | 46.9 | 42.5 |

## D   Visualization Analysis

We visualize the predictions of MaskDINO and **DI-MaskDINO** to show qualitative comparison on BDD100K dataset. As shown in Fig. 3, MaskDINO produces boxes that do not tightly encompass the objects (i.e., Fig. 3a) or do not fully surround the objects (i.e., Fig. 3b). Compared to MaskDINO, our model produces perfectly-fitting boxes, demonstrating the effectiveness of **DI-MaskDINO**. In addition, our model focuses attention on the foreground objects with high category scores through the *residual double-selection* mechanism that avoids mispredicting the background as a foreground object. Fig. 3c suggests that our proposed *residual double-selection* mechanism is effective.

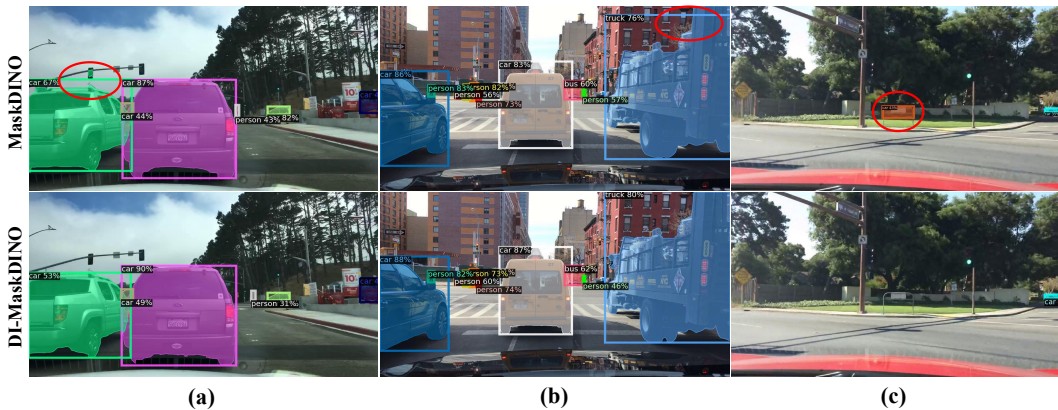

(a)          (b)          (c)

Figure 3: Qualitative comparison between MaskDINO and **DI-MaskDINO** on BDD100K dataset. Suggest zooming in to view this figure for a clearer view of details.

