# OpenReview forum: "DI-MaskDINO: A Joint Object Detection and Instance Segmentation Model"
_NeurIPS.cc/2024/Conference — NeurIPS 2024 poster_

### Official Review · Reviewer_KbFf · 2024-07-08

**Soundness:** 3
**Presentation:** 3
**Contribution:** 2
**Rating:** 5
**Confidence:** 4

**Summary:**

This work investigates the detection-segmentation imbalance issue in MaskDINO. It proposes DI-MaskDINO model with the residual double-selection mechanism to alleviate the imbalance. The framework mainly includes De-Imbalance and Balance-Aware Tokens Optimization. Experiments prove the effectiveness.

**Strengths:**

1. The finding of detection and segmentation imbalance at the beginning of MaskDINO is interesting.
2. Experiments prove the effectiveness on various benchmarks.
3. Overall, the whole framework is clear and easy to follow.

**Weaknesses:**

1. The motivation of De-imbalance module requires more verification. The authors claim "The token interaction is the key point to make sure that the secondly-selected tokens are beneficial for detection" in L169. But there are no experiments or theory to prove this claim or design. This makes the reviewer confused why the token interaction can play such a role. It's better to add some experiments or heatmap visualization.
2. The Balance-Aware Tokens Optimization module contains several components. It's essential to give experimental analysis of each design.
3. This work is built on MaskDINO, it is important to generalize this proposed manner to other decoder-based methods. It can further validate the generality of the proposed approach.
4. Experimental results on COCO test set should be provided for fair comparisons.

**Questions:**

My main concern is the motivation and analysis of the designed component. Please refer to the Weakness section.

**Limitations:**

Limitations are discussed in the work.

---

> ### Author Rebuttal · Authors · 2024-08-07
>
> We are greatly encouraged by your positive comments, including "the finding of detection and segmentation imbalance at the beginning of MaskDINO is interesting" and "the whole framework is clear and easy to follow".
>
> **[W1]** Thank you very much for the insightful observation. To clearer clarify the effect of token interaction, we firstly make the analysis from the theoretical view, and then prove its effect by experiments.
>
> (1) Theoretical analysis
>
> Each token actually corresponds to a patch (remarkably smaller than an object in most cases) in the image [52], and the bounding box of an object is regressed by integrating the multiple patches (belonging to the same object) that have global patch-to-patch spatial relations, thus it is really needed for the detection task to learn the interaction relation between patches. In contrast, the dense all-pixel supervision for the segmentation task mainly focuses on local pixel-level similarity with GT mask [25], hence the segmentation task is not particularly depend on the patch-to-patch relation as the detection task. Via token interaction, different tokens representing the patches (belonging to the same object) can interact with each other to learn the global geometric, contextual, and semantic patch-to-patch relations, benefiting the perception of object bounding boxes. Therefore, executing token interaction makes De-Imbalance module to be more beneficial for detection.
>
> [25] Feng Li, Hao Zhang, Huaizhe Xu, Shilong Liu, Lei Zhang, Lionel M Ni, and Heung-Yeung Shum. Mask dino: Towards a unified transformer-based framework for object detection and segmentation. In Proceedings of the IEEE/CVF Conference on Computer Vision and Pattern Recognition, pages 3041–3050, 2023.
>
> [52] Sixiao Zheng, Jiachen Lu, Hengshuang Zhao, Xiatian Zhu, Zekun Luo, Yabiao Wang, Yanwei Fu, Jianfeng Feng, Tao Xiang, Philip HS Torr, et al. Rethinking semantic segmentation from a sequence-to-sequence perspective with transformers. In Proceedings of the IEEE/CVF Conference on Computer Vision and Pattern Recognition, pages 6881–6890, 2021.
>
> (2) Experimental analysis
>
> To validate the effectiveness of token interaction, we test the performance of our model under two configurations: w/o token interaction and w/ token interaction. The experiment results are reported in Tab.R4 of the rebuttal PDF file. The configuration with token interaction yields higher performance, demonstrating the effect of token interaction.
>
> **[W2]** Thanks for your suggestion. In DI-MaskDINO, De-Imbalance (DI) module is the critical design to alleviate the detection-segmentation imbalance, generating balance-aware query $Q\_{bal}$ in Eq.4. Balance-Aware Tokens Optimization (BATO) module serves as an auxiliary design to utilize $Q\_{bal}$ to optimize the initial feature token $T\_{i}$ and finally compute balance-aware feature tokens $T\_{bal}$. The effect of BATO is validated by the diagnostic experiments on main modules, and the results are reported in Tab.4 of the original version. We can observe that, when BATO is enabled, the performance obtains significant improvement, demonstrating the effectiveness of BATO.
>
> The main component of BATO is the Guiding Token Generation (GTG), which uses $Q\_{bal}$ to generate overall guiding token $T\_{g}$. $T\_{g}$ is then used to optimize $T\_{i}$ via Multi-Head Cross-Attention. Multi-Head Cross-Attention is a fundamental module that could not be disabled (i.e., if it is disabled, $T\_{i}$ will be directly inputted to the transformer decoder as MaskDINO does). Therefore, in the diagnostic experiments on BATO module, we only evaluate the effect of GTG by testing the performance of model under the configurations with/without GTG. The experiment results are reported in Tab.6 of the original version.
>
> **[W3]** Thank you very much for this comment. Based on the structure characteristic of existing transformer-based joint object detection and instance segmentation models, our proposed De-Imbalance module (the core of our model) could be friendly applied on other models. Transformer-based joint object detection and instance segmentation models have a similar model architecture, consisting of backbone, transformer encoder, transformer decoder, and prediction heads. De-Imbalance module takes the output of transformer encoder as the input, and the output of De-Imbalance module is then taken as the input of transformer decoder. Therefore, adding the De-Imbalance module to a existing model will not hurt its main structure. Similarly, our proposed Balance-Aware Tokens Optimization module can also be directly added between De-Imbalance module and transformer decoder. Therefore, our proposed modules can be easily applied to other transformer-based joint object detection and instance segmentation models. We would like to carefully explain why the generality is not prioritized in the original version. Though the generality of a model is important, the prior goal of our work is to make the model achieve SOTA results.
>
> **[W4]** Thank you very much for the suggestion. According to your suggestion, we compare DI-MaskDINO with MaskDINO on COCO test-dev. The results are summarized in Tab.R5 of the rebuttal PDF file, which demonstrates the effectiveness and robustness of DI-MaskDINO. We note that the test-dev evaluation server is only available to object detection task, since the testing needs to upload a prediction results file of json format to the evaluation server provided by the COCO dataset website (https://cocodataset.org/?spm=a2c6h.12873639.article-detail.137.34fa30e89NSdTI#upload), but the website does not provide evaluation server for instance segmentation task. Therefore, the performance of instance segmentation is not reported.

---

> > ### Comment · Reviewer_KbFf · 2024-08-11
> >
> > Thanks for the author's rebuttal. I have read the rebuttal and other reviews. It solved most of my concern. So I'd like to change my rating to Borderline accept.

---

### Official Review · Reviewer_TyXi · 2024-07-08

**Soundness:** 2
**Presentation:** 3
**Contribution:** 3
**Rating:** 5
**Confidence:** 4

**Summary:**

This paper focuses on the detection-segmentation imbalance issue and proposes DI module with the residual double-selection mechanism to alleviate the imbalance; moreover, Balance-Aware Tokens Optimization (BATO) is proposed to guide the optimization of the initial feature tokens. The proposed method termed DI-MaskDINO, achieves SOTA results in both object detection and instance segmentation.

**Strengths:**

1. The authors claim that the performance of object detection lags behind that of instance segmentation from the beginning transformer decoder is interesting;
2. The proposed method achieves SOTA results in both object detection and instance segmentation;
3. The paper is clear, and the experimental results are detailed.

**Weaknesses:**

The technological innovation is somewhat limited.

**Questions:**

The authors implement De-Imbalance module by stacking several self-attention and Multi-Head Cross-Attention. However, it could be seen as one-layer transformer encoder / decoder. The current phenomenon cannot clarify whether the performance improvement stems primarily from mitigating the detection-segmentation imbalance issue or from the increase in parameters/layer. The authors need to provide evidence to support this claim. For example, they could show that a 6-layer DI-MaskDINO outperforms a 7-layer MaskDINO, etc.

**Limitations:**

Yes.

---

> ### Author Rebuttal · Authors · 2024-08-07
>
> We appreciate your positive comments on our work, such as "the proposed method achieves SOTA results" and "the paper is clear and the experimental results are detailed". There are two concerns regarding technological innovation and whether the performance improvement stems primarily from mitigating the detection-segmentation imbalance issue. Our responses to the two concerns are as follows.
>
> **[W1Q1]** "The technological innovation is somewhat limited. De-Imbalance module could be seen as one-layer transformer encoder / decoder."
>
> **Response:** From the technological view, De-Imbalance module is composed of several self-attention and cross-attention. However, we carefully explain that self-attention and cross-attention are basic units to build our De-Imbalance structure. Therefore, **the technological innovations of De-Imbalance module mainly lie in its explanation, simplicity, and effectiveness, which contributes new inspiring thought to the community to push forward the study of fundamental object detection and instance segmentation tasks.** The detailed analyses are as follows.
>
> (1) De-Imbalance module is explainable and effective
>
> In transformer-based joint object detection and instance segmentation models, the two tasks are closely-related and mutually-affected, and the mutual effects could be positive or negative. The imbalance brings the negative impact on model performance. As shown in Fig.1(a) of the original version, at the beginning layer, object bounding boxes do not fit well with object mask, which will hinder the cooperation of the two tasks, leading to the negative effect. In contrast, the interaction under a balanced state makes the two tasks mutually-beneficial.
>
> However, most joint object detection and instance segmentation models (e.g., SOLQ and SOIT) do not focus on how the interaction between the two tasks affects the model performance. We focus on the imbalance issue through studying SOTA joint object detection and instance segmentation model MaskDINO. Furthermore, we propose De-Imbalance module to alleviate the imbalance. It is noted that the detection-segmentation imbalance means that the performance of object detection lags behind that of instance segmentation at the beginning layer of decoder. Therefore, the core idea of De-Imbalance module is to strengthen the performance of detection to alleviate the imbalance at the beginning layer of decoder. By narrowing the imbalance between the two tasks at the beginning layer, the two tasks could rapidly reach a mutually-beneficial state, which contributes to improve the performance. In addition, a large number of experiments (e.g., Tab.3-5 in the original version) have proven the effectiveness of De-Imbalance module.
>
> In summary, De-Imbalance module contribute to handle the long-standing and naturally-existing imbalance issue between object detection and instance segmentation, which is commonly ignored in previous works. Although De-Imbalance module is composed of basic network units (e.g., self attention), they have been proven to be explainable and effective.
>
> (2) The simplicity of our model
>
> To analyze the simplicity of our model, we compare the parameters of DI-MaskDINO configured with different numbers of decoder layers with that of MaskDINO, and the results are summarized in Tab.R3 of the rebuttal PDF file. It is worth noting that our model has only 6 decoder layers (52.3M), while MaskDINO contains 9 decoder layers (52.1M). Our model achieves higher performance, at the cost of involving only 0.2M parameters. We can also observe that our model with 3 decoder layers has achieved comparable performance compared to MaskDINO with 9 decoder layers (i.e., 45.8 v.s. 45.7 on $AP^{box}$ and 41.3 v.s. 41.4 on $AP^{mask}$), saving 4.5M parameters at the same time.
>
> In addition, alleviating the imbalance contributes to accelerate the convergence. As shown in Tab.1 of the original version, DI-MaskDINO presents significant advantage under the training condition of epoch = 12, which potentially reveals that our model reaches the convergence with a faster speed. This is attributed to that alleviating the imbalance issue promotes the collaboration between the two tasks, thus the two tasks could rapidly reach a mutually-beneficial state.
>
> **[Q1]** "Whether the performance improvement stems primarily from mitigating the detection-segmentation imbalance issue or from the increase in parameters/layer."
>
> **Response:** Thank you very much for the insightful comment. According to your comment, we conduct the experiments to validate that the performance improvement does not stem from the increase in parameters/layer. In detail, we test the performance of DI-MaskDINO configured with 3, 6, and 9 decoder layers and compute the parameters of corresponding configurations, respectively. The results are reported in Tab.R3 of the rebuttal PDF file, indicating that the performance improvement does not stem from the increase in parameters or decoder layers. Specifically, our model with 3 decoder layers has achieved comparable performance with MaskDINO, and 4.5M (52.1-47.6) parameters are reduced at the same time. In contrast, the performance improvement stems from mitigating the detection-segmentation imbalance issue, which could be evidenced by the experiment results in Tab.4 of the original version. When enabling De-Imbalance module, the overall performance significantly increases (i.e., from 45.6 to 46.4 on $AP^{box}$ and from 41.2 to 42.1 on $AP^{mask}$).

---

> ### Comment · Reviewer_TyXi · 2024-08-14
>
> The author's rebuttal has solved most of my concern, and I'd like to change my rating to Borderline accept.

---

### Official Review · Reviewer_5ayP · 2024-07-09

**Soundness:** 3
**Presentation:** 3
**Contribution:** 3
**Rating:** 7
**Confidence:** 4

**Summary:**

This paper initially observes that in the current state-of-the-art model MaskDINO, the performance of object detection lags behind instance segmentation at the initial layer of the transformer decoder, resulting in a performance imbalance phenomenon.

To explore whether this "performance imbalance issue" is a factor that restricts the effectiveness of the detector, the authors propose the DI-MaskDINO model, which introduces two key components: the De-Imbalance (DI) module and the Balance-Aware Tokens Optimization (BATO) module, to alleviate the performance imbalance between detection and segmentation tasks.

**Strengths:**

- The paper identifies a novel issue: the performance imbalance that exists between object detection and instance segmentation, a problem that is less discussed in existing literature. This work is of significant importance for advancing research in the fields of object detection and instance segmentation, particularly by providing new perspectives and solutions for dealing with performance imbalance issues.
- The proposed model demonstrates certain improvements over the baseline on the COCO and BDD100K benchmark tests, which to some extent indicates that the "performance imbalance issue" is a factor that restricts the effectiveness of detectors, and the logic of the paper is coherent.

**Weaknesses:**

- The lack of in-depth analysis of "performance imbalance problem" in this paper, only through the experimental phenomenon, there may be a certain coincidence.

**Questions:**

- In Experiment Table 1, why were AP_S, AP_M, and AP_L not tested separately for MaskDINO at epochs 12 and 24?
- In Experiment Table 2, why is there no comparison on the BDD100K validation set for the SwinL backbone?

**Limitations:**

- Although the paper has proposed the "performance imbalance issue," it only conducted experimental observations on MaskDINO. Can other detectors also improve model performance by addressing the "performance imbalance issue"? The paper does not generalize this issue to a more general level, making it more universally applicable.
- The paper's discussion of the "performance imbalance issue" is only focused on the initial layer of the transformer decoder and does not explore whether other layers of the decoder may also have the "performance imbalance issue."

---

> ### Author Rebuttal · Authors · 2024-08-07
>
> **[W1]** We are greatly encouraged by your positive comments "the paper identifies a novel issue, and this work is of significant importance for advancing research in the fields of object detection and instance segmentation, particularly by providing new perspectives and solutions for dealing with performance imbalance issues". The experimental analyses for the "imbalance'' have been conducted, as shown in $\S$ 4.3.1 in the original version. To further solve your concern regarding the "imbalance'', we make the in-depth analysis from the theoretical perspective. The analysis begins with the explanation for the performance imbalance at the beginning layer of transformer decoder, followed by analyzing essential reasons causing the "imbalance''.
>
> The "imbalance'' in our paper is a concise summary for the phenomenon that the performance of object detection lags behind that of instance segmentation at the beginning layer of transformer decoder. The reasons are multi-fold.
>
> Firstly, the individual characteristics and supervision manners of detection and segmentation tasks lead to the "imbalance''. Object detection is a coarse-grained region-level regression task to accurately locate the bounding box of an object, while instance segmentation is a fine-grained pixel-level classification task to correctly classify each pixel of an object. Therefore, object detection relies on more global information that reveals geometric, contextual, and semantic relations of different patches belonging to an object. However, at the beginning layer, the global information is limited, thus the performance of object detection lags behind.
>
> In addition, the supervision for object detection is sparse (i.e., a 4D vector of x, y, w, and h), while the supervision for instance segmentation is dense (i.e., hundreds or thousands dimension vector of GT mask pixels). The sparse supervision is weak, but it encourages a model to learn the relatively global contexts of different patches belonging to an object, which is challenging at the beginning layer. In contrast, the dense supervision is strong, and it is suitable for a model to achieve local pixel-level classification. Therefore, the supervision manners also sharpen the "imbalance'' at the beginning layer.
>
> **[Q1]** Thank your very much for kindly pointing out this detail. We fetch the results from [25] (i.e., MaskDINO) at epochs 12 and 24, and $AP^{box}\_{S}$, $AP^{box}\_{M}$, and $AP^{box}\_{L}$ are not provided in [25]. We are also unable to get the model weights for testing since the weight files are not available.
>
> [25] Feng Li, Hao Zhang, Huaizhe Xu, Shilong Liu, Lei Zhang, Lionel M Ni, and Heung-Yeung Shum. Mask dino: Towards a unified transformer-based framework for object detection and segmentation. In Proceedings of the IEEE/CVF Conference on Computer Vision and Pattern Recognition, pages 3041–3050, 2023.
>
> **[Q2]** Thank you very much for the insightful observation. In the original version, the experiments on BDD100K dataset are treated as additional experiments, thus we only conduct the experiments using ResNet50 backbone on BDD100K dataset. According to your comment, we conducted additional experiments on BDD100K using Swin-L backbone. The results are reported in Tab.R2 of the rebuttal PDF file, from which we can observe that our model presents the advantage.
>
> **[L1L2]** Thank you very much for pointing out the limitations, which inspire us to optimize the model. Regarding the generality, we do not extend the imbalance issue to other models, since the prior goal of our work is to achieve SOTA performance. In the future, we will further study the generality issue. Regarding the imbalance issue in other decoder layers, the imbalance will gradually be weakened with the increasing of decoder layers, thus the imbalance issue is not prominent in other decoder layers.

---

> > ### Comment · Reviewer_5ayP · 2024-08-09
> >
> > Thanks for the authors responce, my concerns are resolved. I rise the score to Accept.

---

### Official Review · Reviewer_aN5C · 2024-07-12

**Soundness:** 3
**Presentation:** 3
**Contribution:** 3
**Rating:** 5
**Confidence:** 4

**Summary:**

The paper starts from an observation regarding imbalance in the intermediate results between detection and instance segmentation, which motivates the authors to propose DI-MaskDINO, which tries to improve the imbalance through the DE-Imbalance module and Balance-Aware Tokens Optimization module. Evaluated on COCO and BDD100K benchmarks, the proposed DI-MaskDINO achieves better results than MaskDINO baseline.

**Strengths:**

- The paper is generally well-written and technically solid

- The starting point of det/seg imbalance sounds interesting

- Extensive experiments and achieve improvement over strong baseline MaskDINO

**Weaknesses:**

- The paper claims improvement over SOTA results, yet it seems that the MaskDINO-SwinL is only reported under 12 epochs setting, while in MaskDINO paper they report results under 50 epochs setting with AP_mask=52.3, AP_box=59.0. I do not see why in tab.1 epochs = 12/24/50 are all reported for R50 backbone but only epochs = 12 are reported for SwinL backbone, making the SOTA claim less comprehensive and less convincing.


- The motivation stems from the observation on det/seg imbalance. Yet, I do not see why performance gap between det and seg from the first layer can illustrate the imbalance. If the absolute AP value is used to measure the imbalance between det and seg, then why AP_box < AP_mask at first layer yet AP_box > AP_mask at the last layer? Furthermore, it is also possible that DI-MaskDINO just improves the performance of both branches which naturally reduce the performance gap (e.g., improving det from 10 to 20 may be in similar difficulty of improving seg from 15 to 22, but the absolute gap is reduced.) In short, I believe there lacks a solid definition and study on the "imbalance" problem, and the current comparison based on absolute AP value from first layer is not convincing to me.


- Minor: I see the reported FPS is significantly different from the ones reported in MaskDINO, which I assume could be the hardware differences. Please explain how the FPS is measured.


- Minor: The paper provides an anonymous link to the code/model, yet the provided repo is empty.

**Questions:**

Though I hold several concerns as illustrated in the weakness, I feel the paper is generally technically good and with performance improvement over strong baseline MaskDINO. Thus my initial rating is boarderline accept. My major concerns lies in the motivation on imbalance is not so convincing to me, I look forward to the author's rebuttal for further illustration or other more convincing measurement.

**Limitations:**

No other limitations as far as I know

---

> ### Author Rebuttal · Authors · 2024-08-07
>
> We are greatly encouraged by your positive comments, including "paper is generally well-written and technically solid" and "the starting point of det/seg imbalance sounds interesting".
>
> **[W1]** We supplement the experiment under the condition of epoch = 50 when the backbone is Swin-L. The results are shown in Tab.R1 of the rebuttal PDF file, indicating that DI-MaskDINO exhibits the superiority over MaskDINO. It is noted that the experiments with the Swin-L backbone are conducted on the same 4 A6000 GPUs with the batch size of 4 (the maximum bacth size that 4 A6000 GPUs supports). The batch size is smaller than that in MaskDINO paper (i.e., batch size = 16) and the 4 A6000 GPUs present weaker computation power than 4 A100 GPUs, thus the results we reproduced are lower than those in the original MaskDINO paper (i.e., $AP^{box}=59.0$, $AP^{mask}=52.3$). In the original version, DI-MaskDINO is compared with MaskDINO under three different settings of epochs = 12/24/50 when the backbone is ResNet, which have demonstrated the superiority of DI-MaskDINO to the certain extent. Therefore, when the backbone is Swin-L, we only test one setting of epoch = 12.
>
> **[W2]** To better answer your question, we firstly explain the definition of "imbalance" and the reason for using the absolute AP to measure the "imbalance". Then, we analyze why $AP^{box}$ is smaller than $AP^{mask}$ at the beginning layer. Finally, we explain why $AP^{box}$ is larger than $AP^{mask}$ at the last layer.
>
> (1) Explanation of "imbalance"
>
> Specifically, the "imbalance'' summarizes the phenomenon that the performance of object detection lags behind that of instance segmentation at the beginning layer of transformer decoder. Therefore, "imbalance'' could be better understood as a concise summary for the above mentioned phenomenon. To quantitatively evaluate the "imbalance'', we previously think about the idea to align $AP^{box}$ and $AP^{mask}$ to (0,1) to better clarify relative AP value. However, it might be controversial to decide the AP value corresponding to 1. If we define that AP=100 is corresponding to 1, then the relative AP equals to absolute AP. Therefore, we directly use the absolute AP to measure the "imbalance".
>
> (2) Why is $AP^{box}$ smaller than $AP^{mask}$ at the beginning layer?
>
> Firstly, the individual characteristics and supervision manners of detection and segmentation tasks lead to the "imbalance''. Object detection is a region-level regression task to locate the bounding box of an object, while instance segmentation is a pixel-level classification task to classify each pixel of an object. Therefore, object detection relies on more global information that reveals the contextual relations of different patches in an object. However, at the beginning layer, the global information is limited, thus the performance of object detection lags behind.
>
> In addition, the supervision for object detection is sparse (i.e., a 4D vector of x, y, w, and h), while the supervision for instance segmentation is dense (i.e., hundreds or thousands dimension vector of GT mask pixels). The sparse supervision is weak, but it encourages a model to learn the relatively global contexts, which is challenging at the beginning layer. In contrast, the dense supervision is strong and suitable for local pixel-level classification.
>
> (3) Why is $AP^{box}$ larger than $AP^{mask}$ at the last layer?
>
> Firstly, as mentioned above, object detection relies on more global information. With the increasing of decoder layers, the global information is richer to benefit object detection.
>
> Secondly, object detection is a coarse-grained task that only needs to locate the four vertices of an object bounding box, while instance segmentation is a fine-grained task that require to accurately classify a large number of pixels of an object mask. Therefore, in an overall view, object detection is relatively simpler. As shown in Tab.1 of the original version, for various models (Mask RCNN, HTC, SOLQ, and MaskDINO), $AP^{box}$ is much higher than $AP^{mask}$, indicating it is easier for object detection to achieve higher final performance. Therefore, the performance of object detection is larger than that of instance segmentation at the last layer.
>
> (4) The significance of our work
>
> The above analysis could potentially reveal the significance of our work. The detection branch and segmentation branch share the unified query, thus the two tasks are interactive and mutually affected. At the beginning layer, as shown in Fig.1(a) in the original version, object bounding boxes do not fit well with object mask, which will hinder the cooperation of the two tasks, leading to the negative interactive effect. By narrowing the imbalance between object detection and instance segmentation at the beginning layer, the two tasks could rapidly reach a mutually-beneficial state, which contributes to improve the performance, as validated by our experiment results in Tab.1 of the original version. In addition, our model also contributes to reduce the parameters of model, as validated by the supplemented experiment results in Tab.R3 of the rebuttal PDF file, which shows that our model configured with 3 decoder layers could achieve the similar performance with MaskDINO configured with 9 decoder layers, and 4.5M (52.1-47.6) parameters are reduced at the same time.
>
> **[W3]** For the fairness, the FPS of all models is tested on the same 4 RTX3090 GPUs when the backbone is ResNet50. When the backbone is Swin-L, the FPS is tested on the same 4 A6000 GPUs, since 4 RTX3090 GPUs could not support the Swin-L backbone.
>
> **[W4]** Despite using the anonymous link, we are still concerned about violating the double-blind reviewing policy. Therefore, we only provide an empty repository to clarify our willingness of releasing the code/model. We will definitely release the code/model if the paper is accepted.

---

### Author Rebuttal · Authors · 2024-08-07

We appreciate the reviewers for their constructive comments and suggestions.

We are particularly encouraged that the reviewers unanimously acknowledge **our work is interesting** (aN5C, 5ayP, TyXi, and KbFf). Reviewers commend us for **achieving state-of-the-art results** (aN5C, 5ayP, TyXi, and KbFf). Reviewers comment positively on **writing and presentation** (aN5C, TyXi, and KbFf). Specifically, reviewer 5ayP remarks that "**the work is of significant importance for advancing research in the fields of object detection and instance segmentation**". Reviewer aN5C notes that our paper is "**technically solid**". Reviewer KbFf finds "**the whole framework is clear and easy to follow**".

---

### Decision · Program_Chairs · 2024-09-25

**Decision:**

Accept (poster)

**Comment:**

This work proposes DI-MaskDINO, aiming to improve the detection-segmentation imbalance of the state-of-the-art model MaskDINO. Specifically, the proposed De-Imbalance (DI) module and Balance-Aware Tokens Optimization (BATO) module are integrated into MaskDINO, where DI generates balance-aware query and BATO uses the query to guide the optimization. Promising results have been demonstrated on COCO and BDD100K datasets.

Initially, the reviewers were concerned about Swin-L experiments, more in-depth analysis of the imbalance problem, and so on. During the reviewer-author discussion period, the author rebuttal successfully assuaged the reviewer concerns. In the end, the paper received all accept recommendations (3 borderline accepts and 1 accept). After considering the author rebuttal and reviewer discussion/comments, the area chair agrees with this recommendation.

The authors are encouraged to incorporate the rebuttal and reviewer suggestions to their final camera-ready version.